# Costs of Newly Funded Proton Therapy Using Time-Driven Activity-Based Costing in The Netherlands

**DOI:** 10.3390/cancers15020516

**Published:** 2023-01-14

**Authors:** Yi Hsuan Chen, Hedwig M. Blommestein, Reinder Klazenga, Carin Uyl-de Groot, Marco van Vulpen

**Affiliations:** 1Erasmus School of Health Policy and Management, Erasmus University Rotterdam, 3062 PA Rotterdam, The Netherlands; 2Holland Proton Therapy Center, 2629 JH Delft, The Netherlands

**Keywords:** proton therapy, cost, time-driven activity-based costing analysis

## Abstract

**Simple Summary:**

Proton therapy delivers more precise treatment compared with conventional radiotherapy. While this innovation entails investment costs, the information about the treatment costs per patient is limited. This information gap might prevent policymakers from making informed decisions. This study aims to calculate the costs of Proton therapy at a single center during the start-up phase and to provide essential information for the health technology assessment of proton therapy. The total cost of proton therapy varied between EUR 12,062 for eye melanoma and EUR 89,716 for head and neck cancer. Overall, indirect costs were the most significant cost component. The high indirect costs implied the potential of the scale of economics; according to our estimation, the treatment cost could be reduced to 35% of the current cost when maximum treatment capacity is achieved. Nevertheless, to have an estimation that reflects the matured cost of proton therapy which could be used in cost-effectiveness analysis, a follow-up study assessing the full-fledged situation is recommended. However, this study provided insights into the financial situation of a new proton therapy center during its ramp-up period and laid the first stone for future costing studies.

**Abstract:**

Background: Proton therapy (PT) has characteristics that enable the sparing of healthy, non-cancerous tissue surrounding the radiotherapy target volume better from radiation doses than conventional radiotherapy for patients with cancer. While this innovation entails investment costs, the information about the treatment costs per patient, especially during the start-up phase, is limited. This study aims to calculate the costs of PT at a single center during the start-up phase in the Netherlands. Methods: The cost of PT per patient was estimated for the treatment indications, head and neck cancer, breast cancer, brain cancer, thorax cancer, chordoma and eye melanoma. A time-driven activity-based costing analysis (TDABC), a methodology that calculates the costs of consumed healthcare resources by a patient, was conducted in a newly established PT center in the Netherlands (HPTC). Both direct (e.g., the human resource costs for medical staff) and indirect costs (e.g., the operating/interest costs, indirect human resource costs and depreciation costs) were included. A scenario analysis was conducted for short-term (2021), middle-term (till 2024) and long-term (after 2024) predicted patient numbers in the PT center. Results: The total cost of PT in 2020 at the center varied between EUR 12,062 for an eye melanoma course and EUR 89,716 for a head and neck course. Overall, indirect costs were the largest cost component. The high indirect costs implied the potential of the scale of economics; according to our estimation, the treatment cost could be reduced to 35% of the current cost when maximum treatment capacity is achieved. Conclusion: This study estimated the PT cost delivered in a newly operated treatment center. Scenario analysis for increased patient numbers revealed the potential for cost reductions. Nevertheless, to have an estimation that reflects the matured cost of PT which could be used in cost-effectiveness analysis, a follow-up study assessing the full-fledged situation is recommended.

## 1. Introduction

Increasing healthcare expenditures have been observed worldwide in the past few decades. The percentage of gross domestic product (GDP) on healthcare increased from 8.6% to 9.9% between 2000 to 2015. In Organization for Economic Co-operation and Development (OECD) countries, the percentage of GDP spent on healthcare increased to 12.5% in 2019 [1]. Of the healthcare expense, the costs related to cancer treatment account for a major portion (6.2%), which requires governments’ attention to constrain their expenditure [2,3].

Around fifty percent of the treatment plans provided to cancer patients include radiotherapy [4], from which external beam radiotherapy (EBRT) is the most widely used in the Netherlands [5]. Despite EBRT’s wide-recognized value in tumor control, its toxicity, caused by radiation damage on healthy tissue, has grown into a reason to develop a more precise alternative.

Proton therapy (PT) is an innovative treatment that offers better precision and aims to tackle the prementioned disadvantages of EBRT [6]. PT damages tumor cells via high-energy ionizing particles while having better control of the depth of tissue penetration [7]. This breakthrough allows doctors to lower the radiation exposure on the healthy tissue near the tumor, which is supposed to largely decrease the incidence of radiation toxicity.

Nevertheless, the treatment advantages of PT come with financial implications. The investment required by PT increased the treatment cost up to three times higher than EBRT [8]. Besides the investment costs, there are also higher operating costs and maintenance fees [9,10]. Additionally, the learning effect plays an important role when introducing an innovative medical device, where its outcome and efficiency are largely related to the experience of the operators [11,12].

PT was first introduced to the Netherlands in 2019, with three PT centers built simultaneously. However, to the best of our knowledge, there is no available micro-costing study based on an operating PT center, not to mention a costing study performed at the early phase of the center being established. The information gap around the cost of setting up an early operating PT center might prevent policymakers from making informed decisions. A suboptimal decision or a delayed decision could lead to damages to societal well-being. This study aimed to estimate the actual incurred costs of PT at a Dutch center and its impact on the healthcare budget during start-up.

## 2. Materials and Methods

### 2.1. Study Design and Sample

This retrospective observational study used data collected from Holland Protonen Therapie Centrum (HPTC), one of the three PT centers in the Netherlands. HPTC is an independent outpatient clinic that provides PT treatment for patients with brain, head and neck, breast, lung, lymphoma, chordoma and eye cancer. HPTC has three treatment rooms, one designed exclusively for eye melanoma, while the other two are identical and applicable for all other indications. Data were collected from all patients receiving PT treatment between January 2020 to February 2021.

### 2.2. Costs

The total cost per patient per indication consisted of two major parts. One was the *direct cost* measured and calculated with the time-driven activity-based costing analysis (TDABC) approach. The other part was the *indirect costs* which included the *indirect human resource costs*, *annual depreciation*, *maintenance* and *overhead cost*.

The *human resource costs* were derived from the corresponding collective labor agreement for University Medical Centers in the Netherlands [13]. The prices of medical consumables, medications and hospital supplies were obtained from the finance records. The *indirect costs* were extracted from the publicly available PT center’s financial statement (2019).

### 2.3. TDABC Method

TDABC is a micro-costing method based on the activities that are provided to the patient that is considered the golden standard for costing studies [14,15]. Unlike its predecessor, activity-based costing, TDABC allocates resource cost directly to a treatment using time in minutes as one unique cost driver [16]. This characteristic provides better transparency and a lower burden in updating care pathway changes to the costing model [17]. Due to these advantages, the TDABC method was applied using the following seven steps [14].

Step 1: Mapping the treatment process

All activities, starting from the patients’ referrals to the PT center up until one year after the last treatment, were registered in the record and verifying (R&V) system of HPTC. The process map constructed and visualized according to the R&V system is shown in Appendix A.

Step 2: Identifying the resource directly involved in each process

The resources involved in the care process were categorized into personnel time (physician, treatment planning technician and nurse) and non-personnel resources (medical consumables, medications and hospital supplies). The personnel time involved in each treatment activity was extracted from the R&V system, in which employees report their working hours. The non-personnel resources involved in each activity were identified by structured interviews.

Step 3: Estimating resources directly involved in each process

Most of the personnel time involved in treatment activities, such as the time the radiation oncologist spent on the appointments, was recorded by the R&V system. For activities that were not recorded by the R&V system, independent structured interviews with a one-month recall period were conducted. To ensure the robustness of this time estimation, multiple employees with the same profession were interviewed. Lastly, the mean personnel time used per patient was calculated by adding up the above estimations.

Step 4: Estimate the capacity and calculate the capacity cost rate (CCR).

After the total time involvement per profession was estimated, the CCR of each profession was calculated. The CCR is the ratio of the cost and the practical capacity of the resource, which is denoted as:$$Capacity\;cost\;rate\;\left(\frac{€}{min}\right)=\frac{cost\;of\;supplied\;capacity}{practical\;capacity\;of\;supplied\;resources}$$

Practical capacity was defined as the number of available working hours per year, accounting for holidays, vacations and leaves. While the cost of human resources was defined as the mean salary of each profession.

Step 5: Calculate the total cost of patient care

The total cost per patient comprised the direct costs and indirect costs. The direct costs per activity were calculated by: (1) multiplying total time involvement per profession by its CCR; (2) adding the cost of non-personnel resource use per activity. The sum of the direct costs of all activities included in the care pathway results in the direct costs per patient (summarized by Equation (1)).

Indirect costs, including indirect human resource, annual depreciation, maintenance, overhead and interest expenditures, were distributed on two schemes. Scheme 1 distributed the indirect costs proportionally to the fraction numbers per course (summarized by Equation (2)). Scheme 2 first distributed indirect costs according to the treatment room size (m^2^), then to patients receiving treatment per treatment room (summarized by Equation (3)). Where one of the three treatment rooms was designated for eye melanoma only, and the other two treatment rooms were available for all other indications. By combining the direct and indirect costs, the total mean costs of a PT course were obtained and summarized by Equation (4).

Equation (1). Direct costs:(1)$$DC_{i}=CCR_{1}\cdot t_{1i}+...\;+CCR_{n}\cdot t_{ni}+O_{i}\cdot p_{i}+C_{i}$$

Equation (2). Indirect costs allocated to each fraction:(2)$$IC_{i\;}=\left(PD+I+R\right)\times \frac{N_{i}\times F_{i}}{\sum_{i}^{m}N_{i}\times F_{i}}$$

Equation (3). Indirect costs allocated to the size of the treatment room:(3)$$IC_{i\;}=\left(PD+I+R\right)\times \frac{G_{i}}{\sum_{1}^{3}G}\times \frac{1}{\left(\sum_{i}^{m}N|G_{i}\right)}$$

Equation (4). Total cost:(4)$$Total\;Cost_{i}=DC_{i}+IC_{i}$$

DC: Direct cost per PT course in EUR,i: Treatment indication,CCR_1→n_: CCR in EUR/minute for human resource group 1, up to group n,t_1→n_: Time involvement in minutes for human resource group 1, up to group n,O: Total cost for optional care pathway activities in EUR,p: Probability of conducting optional care pathway activities in %,C: Costs related to consumables in EUR,IC: Indirect cost per PT course in EUR,D: Depreciation cost in EUR,I: Annual indirect human resource costs in EUR,R: Annual total fixed operating costs in EUR (maintenance, overhead, depreciation, interest),N_i_: Number of treatments delivered for the indication in 2020,F_i_: Mean fraction number per course for the indication.

### 2.4. Statistical Analysis and Outcome

Descriptive statistics, such as gender and age of the study population, were presented in mean and standard deviation. The direct human resource costs per patient per indication were calculated by multiplying the total time involvement per profession by its CCR. The *indirect human resource costs* are proportional to the full-time equivalent (FTE) of non-treatment-related personnel hired in the center (*indirect human resource costs* are equal to *total human resource cost* times the non-treatment-related FTE per total FTE). Twenty-eight percent of the total human resource costs, reported by the annual financial report, were allocated to the indirect human resource cost. The difference between the estimated direct human resource cost and the reported direct human resource cost (financial report) was presented separately (Figure 1). The same scheme was also employed in distributing other indirect costs, including depreciation costs, maintenance costs and other operational costs. The total mean costs per patient per indication were calculated by combining the direct and indirect costs mentioned above.

A probabilistic sensitivity analysis (PSA) was performed using a Monte Carlo simulation with 100,000 iterations to assess the effect of uncertainty in all time estimations and costs. The cost and time were modeled with gamma distributions, which are bound by the interval from 0 to infinity. The standard deviations of costs and time were assumed to be 10% of the mean values.

Scenario analyses were performed to calculate the treatment costs when: (1) the center achieved the expected treatment delivery in 2021 (450 courses per year), (2) the center achieved the maximum treatment capacity as allowed by the government (600 courses per year), (3) the center achieved the full capacity (estimated at 800 courses per year). In all scenario analyses, the indirect costs were distributed to each fraction.

## 3. Results

The descriptive statistics (Table 1) presented the treatment population at HPTC. Roughly two-thirds of the population were females. The mean population age was 49.1 years, and over 75% of the population were between 30 and 64 years old.

Table 2 revealed the total cost of a PT course per treatment indication, with indirect costs distributed to each fraction. The cost per course ranged from EUR 12,062 for eye melanoma to EUR 89,716 for head and neck courses. Whereas, the cost per fraction was around EUR 2500 and was EUR 3015 for eye melanoma. Overall, the operating costs and interest expenditures were responsible for the major part of the cost, around 52% of the total cost. While the direct human resource cost and treatment-related depreciation cost, which accounted for the use of PT systems or CT scanners, took approximately 21% and 11% of the total cost, respectively.

The human resource cost accounted for 66% to 77% of the treatment-related costs. However, with TDABC, only 21% (spinal chordoma) to 59% (eye melanoma) of the direct human resource costs were captured. The human resource costs that occurred during treatment delivery were estimated to range from EUR 1919 (eye melanoma) to EUR 6013 (head and neck cancer) per patient.

Table 3 shows the treatment cost calculated with scheme 2, with which we first distributed the indirect costs proportionally to the treatment room size and then to each patient. The cost per patient ranged from EUR 45,567 (breast cancer) to EUR 137,229 (eye melanoma); and the cost per fraction from EUR 1287 (spinal chordoma) to EUR 34,307 (eye melanoma).

### 3.1. Sensitivity Analysis

To address the uncertainties around the time estimation and costs, a PSA with 100,000 iterations was performed (Table 4). The results from the sensitivity analysis confirmed that the cost per course was within the 20–30% range from the mean value, while the differences between the median and first quantile varied from EUR 340 (eye melanoma) to EUR 3518 (head and neck cancer). The differences between the upper 95% confidence intervals and the mean were within EUR 15 across all indications. These results confirmed the robustness of the cost estimations in this study.

### 3.2. Scenario Analysis

Assuming all other conditions remained the same, the costs per patient per indication were calculated for the following scenarios (Table 5): (1) the projected treatment delivery in 2021, (2) the policy-capped capacity and (3) the full capacity. The course number ranged from 450 to 800 with the same proportion of indications. The scenario analysis showed that, as the number of courses increased, the cost per course would decrease. A 48% to 36% decrease was observed in the 2021 scenario, in which 450 delivered courses were projected. Once the full capacity of the HPTC was achieved, the cost per course would reduce to a third of the current cost in most indications.

## 4. Discussion

This study focused on a new PT center and provided insight into the costs during the start-up phase. The cost per treatment course ranged from EUR 12,062 to EUR 89,716 across different indications (costing detail provided in appendix A). The estimation in this study is higher than the reported price for head and neck cancer (EUR 51k [18]–EUR 40k [19]), skull base chordoma (EUR 31k [19]) and lung cancer (EUR 16k [19]–EUR 28k [20]) but the costs were lower for breast cancer (EUR 34k–EUR 66k [21]), eye cancer (EUR 28k [22]) treatment compared to prior studies. The most expensive indications were head and neck cancer and spinal chordoma. These two indications require more fractions (average 35 to 36 fractions) due to the size, location of the tumor and the complexity of the treatment plan. In this study, all the *indirect costs* and *depreciation costs* were distributed proportionally to the number of fractions per course. Therefore, the costs for these two indications were the highest.

In previous studies, the higher cost was mainly related to high initial capital investment, accompanied by the cost of repaying the loan and interest [9]. Similar findings were observed in our study. In 2019, the interest expenditure alone accounted for 24% of the operation cost in HPTC. Other factors that could have driven up the cost of treatment were the expensive equipment, a labor-intensive treatment process and higher treatment complexity that required an intensively trained staff.

Our results showed that *indirect costs*, which contained *indirect human resource use, overhead cost, depreciation cost* and *interest expenditure,* were the largest cost component. This result aligned with a previous costing analysis for radiotherapy, which was proved to have a large share of the *indirect cost* [23]. However, the percentage of the *indirect cost* estimated in this study (more than 85%) was considerably higher than existing evidence for radiotherapy, which ranged from 56% to 28% [24,25]. This difference might be caused by the single-center setting of HPTC and the low patient number. While an in-house PT center (i.e., located in a hospital) may benefit from sharing *overhead costs*, this is not possible in a single-center setting. In a previous study, the overhead cost of a PT center integrated into a hospital was 67% of the total cost estimated in this study [9].

Several studies focusing on the correlation between the cost of radiotherapy and the facility size suggest treatment costs could be subjected to the economics of scale [26]. The cost per patient could drop by 50% when the patient number rose from 400 to 1600 per year. This price plunge indicated the potential of reduced cost as the patient number increased. There were several reasons for the relatively low number of treatment courses delivered. First, oncologists may require adjustment time for incorporating PT as an option for their patients. Not to mention the process of building a consensus on the use of PT. With the hesitations in the early introduction phase, a low patient number was expected during the first year [10]. In Vanderstraeten et al., a business model was built for setting up a PT center. In the model, 25% of the total patient number at full capacity was anticipated in the first year after start-up. In addition, a four-year ramp-up time was assumed for the total patient number to achieve full capacity. A similar trend was also found in the total patient number in HPTC. After one year of operation, a steady increase in patient number was observed.

The restriction on the number of patients per PT center by the Dutch government, which was implemented for cost control reasons, may also have prohibited its growth. In the Netherlands, PT is fully reimbursed by health insurance for eligible patients. However, due to the remaining uncertainty around the cost-effectiveness of PT, the government capped the maximum patient number per treatment center per year. This restriction was initiated due to concerns regarding the financial impact on the healthcare budget. Nevertheless, a side effect of the limited number of patients allowed to be treated is increased costs per patient. Restricted access might reduce resources being wasted on inefficient products. Yet, the way PT impacted the total healthcare budget essentially differs from other medical innovations (e.g., gene therapy)—the most significant part of the financial impact incurred prior to the first treatment delivery. Our scenario analysis results showed the potential of price per patient to be reduced by 42% if the center delivers treatment to its proposed number in 2021 (450 courses/year). Reimbursement of PT and cost-effectiveness analysis should take this into account.

A considerable gap was observed in *direct human resource cost* when comparing the annual financial report with the estimation based on the recorded time involvement. The gross number of *direct human resource costs* presented in the annual financial report was 3.2 million, while the *direct human resource cost* estimated by the TDABC approach was 0.8 million.

The primary cause of this four-times difference was that only the personnel time spent directly on patient treatment was recorded under our *direct human resource cost*. The personnel time spent on other activities, such as researching and developing new treatment options, was put under *indirect human resource cost*. We assumed seventy per cent of the personnel’s working time would be spent on patient treatment, which was suggested by the guideline for costing studies [27]. In contrast, the working time healthcare providers spent on patient treatment was 22 %, in our results. The assumption that seventy-percent of the time is spent on patient treatment might be too optimistic for facilities that are in a learning curve phase. The reported results were measured in the first year the PT center was funded. For the center’s employees, PT is a new technology that demands time to learn, experiment and optimize all the detailed procedures. The time used to implement the treatment was not included in the treatment delivery pathway and, consequently, lowered the percentage of personnel time spent on direct patient treatment.

Differences in personnel’s salary could be another explanation. In this study, the personnel’s salary was estimated based on the median salary scale derived from the corresponding collective labor agreement. PT requires experienced and high-skilled medical personnel due to the complexity of this advanced technology. This higher requirement could result in a higher salary level.

Another factor that might drive up the *indirect human resource cost* is that the research and development costs partly fall under this group. The PT center is also expected to be a research hub for proton technology. In the center, medical personnel invested their time in research activities, including data collection and experiments. These hours were not captured in the patient-treating pathway and, unless appointed as full-time researchers, fall under the *indirect human resource cost*. We were unable to disentangle these costs.

Although we were dedicated to preserving the granularity rooted in the methodology of TDABC, there was some undeniable uncertainty around the personnel time estimation and salary levels due to data limitations. Estimating the personnel time using the R&V system (e.g., extracting the log-in duration of RTT to a specific patient’s medical file and the time slots of treatment rooms booked under a specific patient) could lead to overestimation in personnel time involved. The uncertainty around salaries is caused by the data limitation. Due to the sensitive nature of this personal information, we could not access the individual salary of each staff member in the treatment center. To address these uncertainties, a PSA was performed with 100,000 iterations. The results of the PSA confirmed that the uncertainties were containable as the difference between the upper 95% confidence intervals and mean within EUR 15 across all indications.

## 5. Conclusions

This study provided a snapshot of the costs of a newly operated PT center and the challenges which could occur in every new treatment center. The results revealed that costs were driven by *indirect costs*, which increased the treatment cost and may ultimately make it more challenging for PT to be cost-effective. However, this study has demonstrated that when increasing the number of treated patients, the costs per patient/treatment course are expected to decrease substantially. Moreover, to have a clear insight into the PT cost for the cost-effectiveness analysis, a follow-up study measuring the full-fledge situation is recommended. With the updated cost information provided by the follow-up studies, the reimbursement price could be adjusted accordingly.

## Figures and Tables

**Figure 1 cancers-15-00516-f001:**
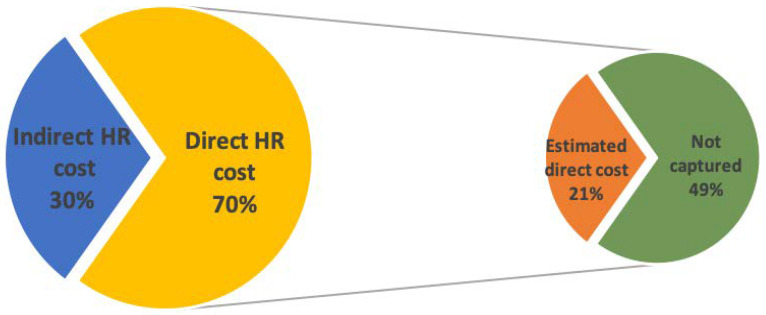
The human resource cost, reported by the annual report and direct human resource cost captured by TDABC.

**Table 1 cancers-15-00516-t001:** Characteristic of the patient population at the PT center.

Patients’ Characteristic	
Gender (Count (%))	
Male	67 (33.7)
Female	132 (66.3)
Age	
Mean	49.1
Age group (count (%))	
18–29	19 (9.5)
30–49	81 (40.1)
50–64	72 (36.2)
65+	27 (13.6)
Cancer type (patient number (average fraction per treatment course))	
Head and neck cancer	30 (35)
Brain cancer	72 (31)
Breast cancer	86 (19)
Thorax cancer	10 (23)
Chordoma (spinal)	8 (36)
Chordoma (skull base)	5 (36)
Eye melanoma	33 (4)

**Table 2 cancers-15-00516-t002:** Average total costs of PT per treatment indication (scheme 1) *.

	Head and Neck	Brain	Breast	Thorax	Chordoma (Spinal)	Chordoma (Skull Base)	Eye Melanoma
Variable costs per patient							
Direct human resource cost	6013	3267	2856	2931	3616	4660	1919
Consumables	430	151	-	69	21	21	144
Other patient-related costs	586	586	586	586	586	586	586
Optional costs (PET/CT)	27	-	-	27	-	-	-
Treatment plan adaptation	300	-	29	-	-	-	-
Uncaptured direct HR cost	14,993	13,066	7925	9639	15,422	15,422	1714
Depreciation costs	10,008	8721	5290	6434	10,294	10,294	1144
Total variable costs	32,357	25,791	16,686	19,685	29,938	30,982	5506
Aggregate fixed costs							
Indirect human resource costs	8551	7452	4520	5497	8795	8795	977
Fixed depreciation costs	2776	2419	1467	1785	2855	2855	317
Operating costs and interest expenditures	46,032	40,113	24,331	29,592	47,347	47,347	5261
Total fixed costs (per patient)	57,359	49,984	30,318	36,873	58,998	58,998	6555
Cost per course	89,716	75,775	47,004	56,559	88,936	85,786	12,062
Cost per fraction	2563	2484	2541	2514	2470	2499	3015

* Estimation based on 244 treatments in 2020 (in euros) with indirect costs distributed proportionally to the number of fractions.

**Table 3 cancers-15-00516-t003:** Average total costs of PT per treatment indication (Scheme 2) *.

	Head and Neck	Brain	Breast	Thorax	Chordoma (Spinal)	Chordoma (Skull Base)	Eye Melanoma
Variable costs per patient							
Direct human resource cost	6013	3267	2856	2931	3616	4660	1919
Consumables	430	151	-	69	21	21	144
Other patient-related costs	586	586	586	586	586	586	586
Optional costs (PET/CT)	27	-	-	27	-	-	-
Treatment plan adaptation	300	-	29	-	-	-	-
Uncaptured direct HR cost	7663	7663	7663	7663	7663	7663	24,500
Depreciation costs	5115	5115	5115	5115	5115	5115	16,353
Total variable costs	20,135	16,783	16,250	16,392	17,002	18,046	43,502
Aggregate fixed costs							
Indirect human resource costs	4371	4371	4371	4371	4371	4371	13,973
Fixed depreciation costs	1419	1419	1419	1419	1419	1419	4536
Operating costs and interest expenditures	23,528	23,528	23,528	23,528	23,528	23,528	75,218
Total fixed costs (per patient)	29,318	29,318	29,318	29,318	29,318	29,318	93,727
Cost per course	49,452	46,100	45,567	45,709	46,319	47,363	137,229
Cost per fraction	1413	1511	2463	2032	1287	1316	34,307

* Estimation based on 244 treatments in 2020 (in euros) with indirect costs distributed proportionally to the size of the treatment room (m^2^).

**Table 4 cancers-15-00516-t004:** Probabilistic sensitivity analysis of the cost per patient per indication based on scheme 1 (Mean (range)).

	Head and Neck	Brain	Breast	Thorax	Chordoma (Spinal)	Chordoma (Skull-Base)	Eye Melanoma
Direct costs per patient							
Human resource cost	6010 (4986–7218)	3268 (2528–4368)	2855 (2302–3505)	2930 (2237–3798)	3616 (2647–4773)	4660 (3652–5896)	1918 (1609–2261)
Other costs	11,269 (11,133–11,434)	9506 (9384–9655)	5857 (5747–6000)	7131 (7016–7284)	10,725 (10,614–10,867)	10,321 (10,209–10,465)	1711 (1589–1858)
Indirect costs							
Human resource costs	24,122 (13,747–37,379)	20,631 (11,757–31,969)	12,401 (7067–19,216)	15,235 (8682–23,608)	23,805 (13,566–36,887)	22,853 (13,023–35,412)	2308 (1315–3577)
Other costs	50,007 (31,848–76,074)	42,769 (27,238–45,419)	25,707 (16,372–39,108)	31,583 (20,115–48,047)	49,349 (31,429–75,073)	47,375 (30,172–72,070)	4785 (3048–7280)
Total costs	91,409 (69,719–118,863)	76,174 (57,544–99,357)	46,820 (35,600–60,847)	56,880 (43,176–74,274)	87,496 (65,773–114,601)	85,208 (64,374–111,191)	10,723 (8541–13,471)

**Table 5 cancers-15-00516-t005:** Costs per PT course with short-term (2021), middle-term (till 2024) and long-term (after 2024) projections (in euros) *.

	Head and Neck	Brain	Breast	Thorax	Chordoma (Spinal)	Chordoma (Skull Base)	Eye Melanoma
Annual Patients: 244 (2020)							
Patient number	30	72	86	10	8	5	33
PT	89,716	75,775	47,004	56,559	88,936	89,980	12,062
Annual Patients: 450 (2021)							
Patient number	55	133	159	18	15	9	61
PT	52,013	42,920	27,076	32,321	50,156	47,006	7753
Annual Patients: 600 (Capped capacity till 2024)							
Patient number	74	177	211	25	20	12	81
PT	40,849	33,191	21,174	25,144	38,673	35,523	6477
Annual Patients: 800 (Full capacity)							
Patient number	98	236	282	33	26	16	108
PT	32,476	25,894	16,749	19,761	30,060	31,104	5520

* Costs calculated with scheme 1; all the indirect costs were distributed to each fraction.

## Data Availability

The data presented in this study are available in this article.

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
