# Peer review of "Costs of Newly Funded Proton Therapy Using Time-Driven Activity-Based Costing in The Netherlands"

_cancers, 2023, doi:10.3390/cancers15020516_

Round 1

Reviewer 1 Report

This manuscript dealt with the cost related to proton therapy unit operation, which most physicians may not be familiar with but could be practically very important issue, when considering the investment level.

Authors’ way of calculating the overall cost seems reasonable, however, not easy to be understood by most physicians.

It is advised for the authors to rephrase their sentences using more easily understandable terminology.

At a glance, the direct cost in treating head and neck cancer patient was the highest, and it is not quite easy to catch up the reason for this. Authors are advised to provide detail information (ex: minutes needed for planning, number of dedicated personnel on each subsites).

No information or suggestion on the reimbursement level depending on the calculated cost is shown throughout the manuscript.

Conclusion (“snapshot”) seems too simple.

Improvement in English expressions are highly recommended.

Author Response

Dear reviewer,

We thank you for taking the time to review our manuscript and for the helpful feedback.

We tried to give more explanation to improve the understandability of our study by physicians and adjusted the manuscript accordingly (detail in the attachment).

Best regards,

Yi Hsuan Chen

Reviewer 2 Report

The manuscript “Costs of Newly Funded Proton Therapy Using Time-Driven Activity-Based Costing in The Netherlands” is addressing a very important issue; what is the cost of proton therapy in the perspective of more or less constantly increasing health care spendings?

The manuscript is very well written and the aim of the study clearly described at the end of the Introduction. Also, the study is exemplary set in perspective by  the authors (in the Introduction and in particular the last two paragraphs).

I am not a health economy specialist, but the methods described seem to be very solid and robust. It is interesting once again to see that the high initial capital investments is very important for the total cost; about 1/4 of the cost in this study. There are also a lot of other interesting cost data in the manuscript which I find publishable. However, it is necessary to have a health economy specialist evaluate the methodology used in the manuscript as this is an area I am not familiar with.

Minor issues:

I find the first sentence in the Abstract a but imprecise: “Proton beam therapy (PBT) may deliver more precise treatment compared with conventional radiotherapy…”. Conventional radiotherapy – for all practical purposes photon radiotherapy – is most often very precise, however, has other characteristics/properties than proton radiotherapy which make sparing of healthy, non-cancerous tissue hard. I suggest to rephrase to “Proton beam therapy (PBT) has characteristics which enable sparing of healthy, non-cancerous tissue surrounding the radiotherapy target volume better from radiation dose than conventional radiotherapy…”.

Also, I think the following sentence in the Introduction “This breakthrough enabled doctors lowering the radiation exposure on the healthy tissue near the tumor, which largely decreases the incidence of irradiation toxicity.” is a bit too bombastic, and I suggest to substitute “largely decreases” with “is supposed to largely decrease”.

Author Response

Dear reviewer,

Thank you for the helpful feedback and for taking the time to review our manuscript.

We tried to give more explanations to improve the understandability of our study and adjusted the manuscript accordingly (detail in the attachment).

Best regard,

Yi Hsuan Chen

Round 2

Reviewer 1 Report

Authors' effort to improve the understandibility has been appreciated.

However, there are still many spots where English expressions are ackward and that need to be improved further.

For example, "proton beam therapy", "proton therapy", "proton treatment center", "proton center" are mixedly used throughout the manuscript.

"micro costing" may be changed into "micro-costing".

There are several spots with grammatically errors and similar words are repeatedly used.

Author Response

Dear reviewer,

We thank you for taking the time to review our manuscript again and for the helpful feedback.

We adjusted the English expressions accordingly (detail as follows).

Point 1: "proton beam therapy", "proton therapy", "proton treatment center", "proton center" are mixedly used throughout the manuscript.

Response 1:

All terms are changed to “proton therapy(PT)” and “proton therapy (PT) center”.

Point 2: "micro costing" may be changed into "micro-costing".

Response 2:

All “micro costing” are changed to “micro-costing”.

Point 3: There are several spots with grammatically errors and similar words are repeatedly used.

Response 3:

Line 43 and 226: The sentences were revised to enhance readability.

Several changes were made to correct the grammar errors and to replace similar words.

Best regards,

Yi Hsuan Chen